# Chloroquine Downregulation of Intestinal Autophagy to Alleviate Biological Stress in Early-Weaned Piglets

**DOI:** 10.3390/ani10020290

**Published:** 2020-02-12

**Authors:** Simeng Liao, Shengguo Tang, Meinan Chang, Ming Qi, Jianjun Li, Bie Tan, Qian Gao, Shuo Zhang, Xiaozhen Li, Yulong Yin, Peng Sun, Yulong Tang

**Affiliations:** 1Laboratory of Animal Nutritional Physiology and Metabolic Process, Key Laboratory of Agro-ecological Processes in Subtropical Region, National Engineering Laboratory for Pollution Control and Waste Utilization in Livestock and Poultry Production, Institute of Subtropical Agriculture, Chinese Academy of Sciences, Changsha 410125, China; lsm19931@163.com (S.L.); tangshengguo1983@hunau.edu.cn (S.T.); qmcharisma@sina.com (M.Q.); jianjunli@isa.ac.cn (J.L.); bietan@isa.ac.cn (B.T.); yinyulong@isa.ac.cn (Y.Y.); 2State Key Laboratory of Animal Nutrition, Institute of Animal Sciences, Chinese Academy of Agricultural Sciences, Beijing 100093, China; meinanchang2020@163.com; 3College of Advanced Agricultural Sciences, University of Chinese Academy of Sciences, Beijing 100008, China; 4College of Animal Science and Technology, Hunan Agricultural University, Changsha 410128, China; gaoqian996@163.com; 5Yunnan Yin Yulong Academician Workstation at Yunnan, Yin Yulong Academician Workstation, Yunnan Xinan Tianyou Animal Husbandry Technology co. Ltd., Kunming 650032, China; jschen@hunan.edu.cn (S.Z.); lxzp888@163.com (X.L.)

**Keywords:** autophagy, chloroquine, mucosal barrier, weaning stress, rapamycin

## Abstract

**Simple Summary:**

Weaning is one of the biggest challenges in a pig’s life. Autophagy is a catabolic process aimed at recycling cellular components and damaged organelles in response to diverse stress conditions. There are two autophagy-modifying agents, rapamycin (RAPA) and chloroquine (CQ), that are often used in vitro and in vivo to regulate this process. We speculated that the regulation of autophagy may have some effect on weaning pressure. In this study, we try to understand the role of autophagy in intestinal barrier function and inflammation during the first week after weaning. We examined the effects of modulation of autophagy via RAPA and CQ on growth performance, immunity, inflammation profile, and the intestinal barrier to find potential value for CQ as a feed additive agent for ameliorating weaning stress.

**Abstract:**

Early weaning stress impairs the development of gastrointestinal barrier function, causing immune system dysfunctions, reduction in feed intake, and growth retardation. Autophagy was hypothesized to be a key underlying cellular process in these dysfunctions. We conjectured that rapamycin (RAPA) and chloroquine (CQ), as two autophagy-modifying agents, regulate the autophagy process and may produce deleterious or beneficial effects on intestinal health and growth. To explore the effect of autophagy on early weaning stress in piglets, 18 early-weaned piglets were assigned to three treatments (each treatment of six piglets) and treated with an equal volume of RAPA, CQ, or saline. The degree of autophagy and serum concentrations of immunoglobulins and cytokines, as well as intestinal morphology and tight junction protein expression, were evaluated. Compared with the control treatment, RAPA-treated piglets exhibited activated autophagy and had decreased final body weight (BW) and average daily gain (ADG) (*p* < 0.05), impaired intestinal morphology and tight junction function, and higher inflammatory responses. The CQ-treated piglets showed higher final BW, ADG, jejuna and ileal villus height, and lower autophagy and inflammation, compared with control piglets (*p* < 0.05). Throughout the experiment, CQ treatment was beneficial to alleviate early weaning stress and intestinal and immune system dysfunction.

## 1. Introduction

The process of weaning is one of the most stressful events in a pig’s life. It involves exposure to physiological, environmental, and social challenges, leading to intestinal and immune system dysfunctions, which reduces feed intake, thereby producing deleterious effects on overall growth and health [1,2,3]. It is well known that maturation of the gastrointestinal epithelial and immune system is a gradual process that approaches completion at around 10 to 12 weeks. However, in commercial pig production, weaning is abrupt, occurring early (between 14 and 30 days of age) [3]. The piglet must adapt abruptly from a liquid milk diet, which is highly digestible and palatable, to a solid dry diet, which is considerably less digestible and palatable. This change results in anatomical and physiological changes to the intestine, thus further affecting intestinal barrier function and absorptive capacity [4,5]. With the disruption of the intestinal barrier, toxins, bacteria, and feed-associated antigens can cross the epithelium, and pro-inflammatory cytokines are activated, then resulting in malabsorption, diarrhea, and poor growth performance [2,6].

Autophagy is a catabolic process aimed at recycling cellular components and damaged organelles in response to diverse stress conditions [7,8], such as nutrient deprivation, viral infection, and genotoxic stress. It is upregulated by cellular stressors, to enhance cell survival [9,10]. Our previous studies demonstrated that autophagy plays a crucial role in maintaining intestinal epithelial cell homeostasis under deoxynivalenol toxin stress [11]. Loss of autophagy, induced by knockout of the ATG5 gene, heightened stress reactions via the generation of reactive oxygen species (ROS) [10]. It has also recently been revealed that the autophagy pathway and its proteins have a central role in controlling other aspects of immunity and inflammation. They keep a balance of the beneficial and detrimental effects on immunity and inflammation, and thereby protect against infectious, autoimmune, and inflammatory diseases [12,13,14].

It is therefore of interest, in the context of piglet weaning, to understand the role of autophagy in intestinal barrier function and inflammatory response during the first week after weaning. Rapamycin (RAPA) and chloroquine (CQ), autophagy-modifying agents, are often used in vitro and in vivo to regulate this process [15,16]. RAPA, a specific inhibitor of the mechanistic target of the rapamycin (mTOR) signaling pathway, can enhance autophagy by inducing the microtubule-associated protein’s light chain 3 (LC3) flux in vivo and in vitro [17]. It has been demonstrated that treating zebrafish embryos by RAPA severely affects the development of the digestive tract, resulting in developmental delay [18]. Chloroquine (CQ) and its derivatives have been widely used to inhibit autophagy in vitro with the benefit of relatively low toxicity. CQ, a lysosomal inhibitor, has been shown to reverse autophagy by accumulating in lysosomes and disturbing the vacuolar H^+^ATPase; this enzyme is responsible for lysosomal acidification and blocking autophagy in cell and animal models [16]. In the dextran sulfate sodium (DSS)-induced murine colitis model, CQ administration significantly retarded colon length shortening, inflammatory cell infiltration, tissue damage, and body weight loss [19]. Based on previous study, we hypothesized that RAPA-overactivated autophagy produced deleterious effects on intestinal health and growth, while CQ partially inhibited autophagy that would ameliorate negative effects of weaned stress and improve production efficiency. In this study, the effects of modulation of autophagy via these two agents on the growth performance, immunity, inflammation profile, and the intestinal barrier were determined after the 14-day postweaning experiment. Thus we were able to assess their potential value as feed supplements for ameliorating weaning stress.

## 2. Materials and Methods

The experimental design and procedures used in this study were approved by the Animal Care and Use Committee of the Institute of Subtropical Agriculture, Chinese Academy of Science (IACUC#201302) approved on 20180421. The RAPA and CQ components (purity ≥98%) were obtained from Sangon Biotech Co., Ltd. (Shanghai, China).

### 2.1. Animals Experimental Design, and Diets

A total of 18 Large White × Landrace piglets, weaned at 24 days (d) of age (average initial body weight (BW) of 7.03 ± 0.57 kg), were assigned into the following three treatments: control, RAPA (1 mg/kg BW) [2], and CQ (10 mg/kg BW) [20,21]. Each treatment had 6 replicates with 1 pig per replicate and there was no significant difference in average BW among three treatments. Piglets were orally administrated with 7 mL saline, RAPA, or CQ once a day on d 27 of age. On the seventh day of the experimentation (on d 33 of age), the total volume of solution was adjusted according to the average weight of each treatment. The piglets were individually housed in cages equipped with a feeder for ingestion and a nipple drinker for free access to drinking water. The basal diet was formulated to meet the nutrient requirements for weanling piglets (NRC, 2012) and was presented in a previous study [22]. Before the formal experiment, piglets were allowed to adapt for 3 days, then the experiment lasted 14 d.

### 2.2. Sample Collection and Preparation

All piglets were weighed individually at the beginning and the end of the experiment. The feed consumption was recorded and weight gain of the piglets was measured at the end of the trial. Those data were used for calculating average daily gain (ADG), average daily feed intake (ADFI), and the gain-to-feed ratio (GF). On d 40 of age, piglets were euthanized and 10 mL of blood sample were collected via jugular vein puncture into two 5 mL tubes for the determination of serum and plasma metabolites. The samples were centrifuged at 3000× *g* and 4 °C for 10 min to separate out the serum and plasma, and samples were stored at −20 °C until analysis, as previously described [23]. All of the animals were humanely euthanized by a lethal intraperitoneal injection of sodium pentobarbital [24]. After euthanasia, jejunal and ileal samples (2 cm, jejunum as the 1/3 mid and ileum as 1/3 distal part) were collected for the determination of intestinal morphology. Then, samples for histological slicing were rapidly fixed with 10% neutral buffered formalin. An approximate 0.5 cm sample of the jejunum and ileum were immediately and rapidly excised with ice-cold physiological saline [22], then stored in form of formaldehyde solution or at 2.5% glutaraldehyde solution until further analysis.

### 2.3. Western Blotting Analysis

Relative protein levels of Beclin1, Sequestosome 1 (P62 /SQSTM1, P62), LC3, and β-actin in the jejunum were determined by Western blotting, as described previously [5]. The primary antibodies used in the present study were as follows: anti-Beclin1 (#3495), anti-P62 (#23214), anti-LC3B, and anti-β-actin (#4970) (Cell Signaling Technology Co., Ltd., Danvers, MA, USA). The second antibody, Goat Anti-Rabbit IgG H&L (ab150077), was purchased from Abcam (Shanghai, China). Chemiluminescent reagent (BeyoECL Plus, Beyotime, Shanghai, China) with a ChemiDoc™ Touch Imaging System (Bio-Rad, Philadelphia, PA, USA) was used to visualize the bands of the protein. The resultant signals were quantified as described previously [25].

### 2.4. Transmission Electron Microscope

An electron microscope was used to observe the autophagosomes in enterocyte cells to note the initiation of autophagy. To assess autophagic vacuoles in the jejunal cells, the jejunal tissue was cut into small pieces (approximate 1 mm^3^) and immediately fixed in 2.5% glutaraldehyde solution at 4 °C overnight. After being postfixed in 1% osmic acid and dehydrated with step-wise gradient ethanol (30, 50, 70, 80, 95, 100%), the samples were embedded in an epoxy resin. Samples then were cut on an LKB-NOVA ultramicrotome into 70 nm sections, which were examined under a HITACHI-600IV electron microscope (HITACHI, Tokyo, Japan)

### 2.5. Serum Inflammatory Cytokines, Immune Factors, Diamine Oxidase, and D-lactate

Serum concentrations of immunoglobulin (Ig) G, IgM (IgG and IgM quantitation kit; Bethyl Laboratories, Inc., Montgomery, TX, USA), tumor necrosis factor (TNF)-α, interferon (IFN)-γ, interleukin (IL)- 1β, IL-6, IL-8, IL-10, IL-12, and transforming growth factor (TGF)-β were determined using ELISA kits (Cell Biolabs, San Diego, CA, USA), according to the manufacturer’s instructions, as previously reported [26].

Serum diamine oxidase (DAO) content was analyzed by a UV/visible spectrophotometer-UV-2450 (Shimadzu, Kyoto, Japan) according to the previous study [27,28]. D-lactate was determined using a D-lactate assay kit (Bio Vision, Mountain View, San Francisco, CA, USA) according to the manufacturer’s instruction.

### 2.6. Plasma Antioxidative Capacity

Determination of superoxide dismutase (SOD), malondialdehyde (MDA), glutathione S-transferase (GST), glutathione peroxidase (GSH-Px), and total antioxidant capacity (T-AOC) levels in plasma were measured by spectrophotometric methods using a UV/visible spectrophotometer (UV-2450, Shimadzu, Kyoto, Japan) according to manufacturer instructions of assay kits (Nanjing Jiancheng, Nanjing, China).

### 2.7. Intestinal Morphology

After 48 h of fixation, the sections of intestinal tissues stored in 10% neutral buffered formalin were washed, excised, dehydrated, and embedded in paraffin wax, and then five transverse sections were sliced, installed on glass slides, and stained with hematoxylin and eosin (H and E) for morphology analysis. Villous height, crypt depth, intestine wall thickness, goblet cell, and lymphocyte were measured and counted with computer-assisted microscopy (Micrometrics TM; Nikon ECLIPSE E200, Tokyo, Japan). At least 20 orientated villi and their adjoining crypts were selected randomly on each slice and the ratio of villus height to crypt depth was calculated and used for further analysis.

### 2.8. Real-Time Quantitative Reverse Transcriptase PCR

Total RNA from the jejunum and ileum was extracted using Trizol reagent (Invitrogen, Thermo Fisher Scientific, USA) according to the manufacturer’s instructions. Purified extracted RNA was obtained by using an RNeasy Mini Kit (Takara Bio Inc., Shiga, Japan), and treated with DNase (Takara, Shuzo, Kyoto, Japan) to remove contaminants. The quality and quantity of RNA were determined by ultraviolet spectroscopy using a Nanodrop 2000 Spectrophotometer (Thermo Scientific, Courtaboeuf, France). With optical densities (ODs) measured at 260 nm, a ratio of OD 260 to OD 280 between 1.8 and 2.0 was considered acceptable [28]. In addition, RNA integrity was verified by gel electrophoresis stained with ethidium bromide (Egel, Invitro-gen, Inc.). It is considered acceptable when there are three bands: 5s RNA, 18s RNA, and 28s RNA. Construction of the cDNA library used the TransScript One-Step gDNA Removal and cDNA Synthesis SuperMix kit (Transgen, Beijing, China), then diluted to 1:20 for the subsequent experiments. Amplification conditions were performed as follows: (1) denaturation for 10 min at 95 °C, (2) 40 PCR cycles of denaturation for 15 s at 95 °C, annealing for 15s at 56–64 °C, and extension for 45s at 56–64 °C.

All PCR primers used in this study are listed in Table 1. Quantitative real-time PCR (qRT-PCR) was performed in triplicate for each complementary DNA (cDNA) sample on an ABI7900HT real-time PCR system (Applied Biosystems, Forrest City, CA, USA) using SYBR^®^ Green I (SYBR^®^ Premix Ex TaqTM II, Takara Biotechnology Co. Ltd., Dalian, China) as a PCR core reagent in a final volume of 10 µL. All procedures were carried out in accordance with the manufacturer’s instructions. Relative expression of the target gene was normalized to β-actin (internal reference) and determined by the 2^–∆∆Ct^ method [28].

### 2.9. Statistical Analysis

Data were analyzed by one-way ANOVAS that compared control vs. RAPA and control vs. CQ, but not RAPA vs. CQ, using the general linear models procedure of the SPSS 20.0 (SPSS Inc., Chicago, IL, USA). Significant differences between means were determined using Tukey’s multiple comparison tests. Results are expressed as the mean ± standard error of mean (SEM) in figures, while results are expressed as the mean, total standard error of mean, and *p*-value in tables. A value of *p* < 0.05 was considered statistically significant, whereas 0.05 ≤ *p* < 0.1 was considered a trend.

## 3. Results

### 3.1. Growth Performance

Compared with the control, orally administered RAPA decreased the final BW by 28% and decreased ADG, ADFI, and GF (*p* < 0.05). In contrast, CQ treatment increased BW by 4.4%, and significantly increased ADG and GF (*p* < 0.05) (Table 2).

### 3.2. Autophagy Analysis

The CQ treatment showed indices at levels comparable to those of the control treatment, while RAPA-treated piglets displayed more autolysosomes and autophagosomes, suggesting higher autophagic activity in this treatment (Figure 1).

Consistent with the investigation of the results under the transmission electron microscope, the piglets supplemented with RAPA showed a significant increase in beclin1 and the proportion of LC3II/LC3I protein, as well as a decrease in the expression of the P62 protein, compared to the control treatment (*p* < 0.05). In contrast, the CQ-treated piglets showed decreased beclin1 protein levels and increased expression levels of P62 protein when compared with the control treatment (*p* < 0.05), indicating inhibition of autophagy.

### 3.3. Intestinal Morphology and Permeability

The evaluated intestinal morphologic parameters included the villus height, crypt depth, ratio of villus height to crypt depth, and wall thickness (Figure 2). The RAPA treatment showed significantly lower levels of villus height and crypt depth when compared to the control treatment (*p* < 0.05), while higher villus height and the ratio of villus height to crypt depth in the jejunum of CQ-treated piglets were observed (*p* < 0.05).

The levels of DAO and D-lactate in serum are markers of intestinal permeability (Table 3). Compared with the control treatment, the RAPA treatment had higher concentrations of D-lactate and levels of DAO in serum (*p* < 0.05), while the CQ treatment had significantly reduced serum D-lactate concentration (*p* < 0.05).

In addition, intestinal permeability was associated with the levels of tight junction proteins (Figure 3). Compared to the control treatment, the RAPA treatment had lower mRNA levels for *E-cadherin, occludin, ZO-1*, and *integrin* in the jejunal and ileal mucosae (*p* < 0.05). The CQ-treated piglets exhibited higher expression levels of *E-cadherin* and *integrin* in the ileal mucosa than in the control treatment (*p* < 0.05).

### 3.4. Inflammatory Cytokine, Immune, and Antioxidant Factor Profiles

Compared with the control treatment, the weaning piglets exposed to RAPA showed upregulated (*p* < 0.05) pro-inflammatory cytokines IL-12, IL-8, TNF-α, IL-6, and IL-1β, but downregulated (*p* < 0.05) anti-inflammatory cytokine TGF-β as well as IgG and IgM in serum; the CQ piglets’ inflammatory status was indicated by lower serum concentrations of IL-1β, IL-8, and IFN-γ (*p* < 0.05) (Figure 4 and Table 4).

In agreement with the results of inflammation, RAPA-treated piglets exhibited remarkable decrease in serum concentrations of SOD, GSH-Px, T-AOC, and GST (*p* < 0.05), as well as an increase of MDA (*p* < 0.05). However, CQ-treated piglets showed higher serum concentrations of GSH-Px and GST compared to the control treatment (*p* < 0.05) (Table 5).

## 4. Discussion

Investigations are currently underway to develop effective methods to minimize adverse effects of weaning stress. We explored the role of autophagy in regulating weaning stress by using two autophagy-modifying agents, RAPA and CQ [16], which are often used in vivo and in vitro, to modulate autophagy. We found that RAPA activated autophagy, producing deleterious effects on growth and health. In contrast, CQ inhibited autophagy, ameliorating negative effects and improving production efficiency.

To further investigate the underlying mechanisms, we evaluated the intestinal function and inflammatory response. In the intestinal mucosa, morphology and tight junction status reflect digestion and absorption of nutrients as well as permeability [29]. The RAPA-treated piglets exhibited significantly impaired permeability and nutrient transport, resulting in reduction of growth performance. Compared to the control piglets, the addition of CQ during the early-weaning period ameliorated intestinal dysfunction as indicated by villus height and tight junction protein expression. Beyond the analysis of tight junction proteins, levels of another two intestinal permeability indicators [2] were recorded. D-lactate is a product released by many microflorae residing in the gastrointestinal tract and DAO is synthesized only via intestinal villi [30]. The RAPA treatment had significantly higher permeability, whereas the CQ-treated piglets had lower levels of D-lactate and better barrier function. These results indicate that piglets experienced excessive autophagy during the weaning period due to RAPA supplement-induced deleterious effects on intestinal development [19,31]. This may be associated with autophagy-dependent cell death in intestinal epithelial cells, known as type 2 cell death [13,32]. Conversely, decreased autophagy, following supplementation with CQ, was beneficial to intestinal morphology and barrier function [29].

We conclude that increased intestinal permeability indicates impaired barrier function and that increased uncontrolled passage of substances through the barrier causes increased inflammatory responses [33]. Indeed, RAPA-treated piglets with disrupted intestinal integrity showed higher levels of pro-inflammatory cytokines, such as IL-1β, IL-6, and TNF-α, compared to the control treatment. Addition of CQ slightly alleviated the inflammatory response, with lower pro-inflammatory cytokines. Meanwhile, inflammatory cells release a number of ROS at the site of inflammation, leading to aggravation of oxidative stress, in turn, its product also enhances the pro-inflammatory response [11]. To alleviate the damage caused by oxidative stress, the host forms the first line of the defense system, composed of antioxidant enzymes, such as SOD and GSH-Px, against the deleterious effect of free radicals [5]. Higher antioxidant contents may contribute to beneficial health effects on anti-inflammation and oxidative stress [4]. Higher levels of antioxidants in serum of CQ-treated piglets indicated that the reduced inflammation was associated with better antioxidant status. CQ has also been extensively used for decades as a drug to mediate autophagy [16]. Alongside its antimalarial activity, it may affect multiple cellular processes, including cellular stress response, antigen presentation, and oxidative stress response [34]. These effects may contribute to the alleviation of the weaning stress response in piglets. The results from previous studies showed CQ promoted injury recovery by preventing the degradation of the autophagic and ubiquitinated protein and inhibiting autophagy-associated inflammation [33,35].

## 5. Conclusions

In conclusion, the inhibition of autophagy by CQ can help weaned piglets maintain intestinal mucosal homeostasis via improving intestinal morphology and barrier function, and decreasing pro-inflammatory cytokines, which produces better antioxidant status. These results support the hypothesis that CQ partially inhibited autophagy. This would ameliorate the negative effects of weaned stress, which provide references for the regulation of ameliorating early weaning stress in piglets.

## Figures and Tables

**Figure 1 animals-10-00290-f001:**
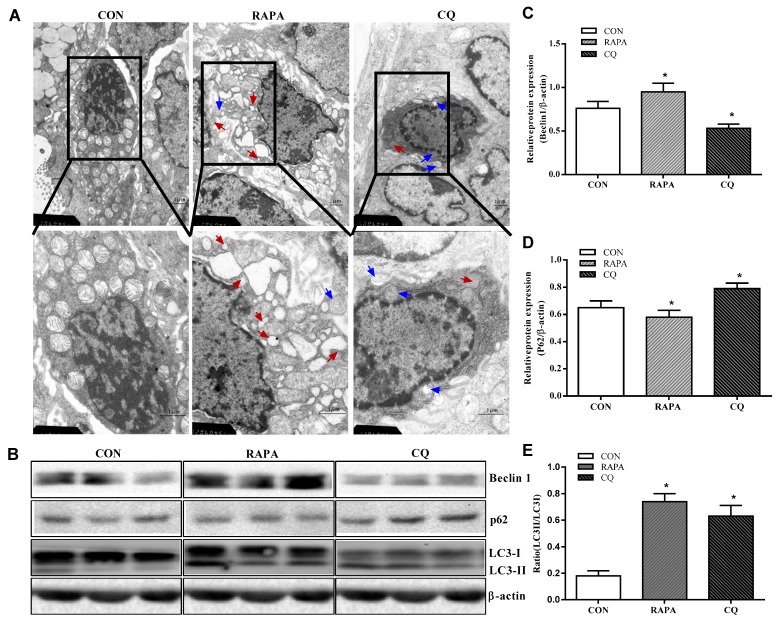
Autophagy validation in the jejunum of weaning piglets. (**A**) Observation of autophagy bubbles by transmission electron microscopy. Red arrows indicate autolysosomes, blue arrows indicate autophagosome. (**B**) Western blot results. (**C**) Beclin1 protein abundance. (**D**) p62 protein abundance. (**E**) LC3B protein abundance (the ratio of LC3II and LC3I). Dietary treatment: RAPA, rapamycin; CQ, chloroquine; CON, normal saline. Data are expressed as means ± SEM (*n* = 6), three models chosen, all values are expressed in relation to the control. * *p* < 0.05 vs. control.

**Figure 2 animals-10-00290-f002:**
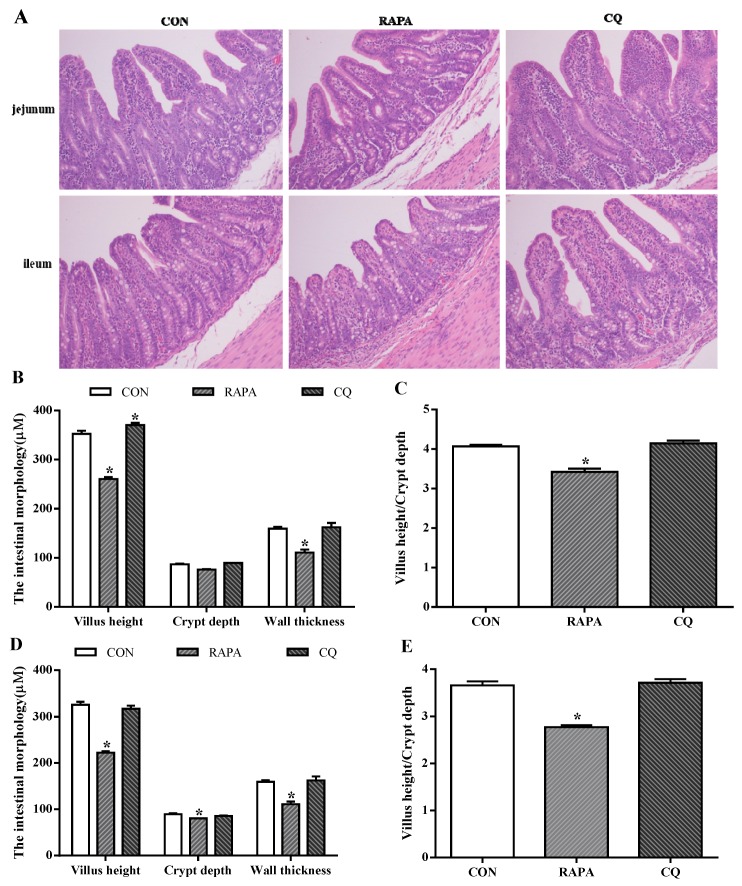
Morphology analysis. (**A**) Sections were stained with hematoxylin and eosin (H and E). (**B**) Jejunum morphology. (**C**) Ratio of villus height to crypt depth in jejunum. (**D**) Ileum morphology. (**E**) Ratio of villus height to crypt depth in ileum. Dietary treatment: RAPA, rapamycin; CQ, chloroquine; CON, normal saline. Data are expressed as means ± SEM (*n* = 6), all values are expressed in relation to the control. * *p* < 0.05 vs. control.

**Figure 3 animals-10-00290-f003:**
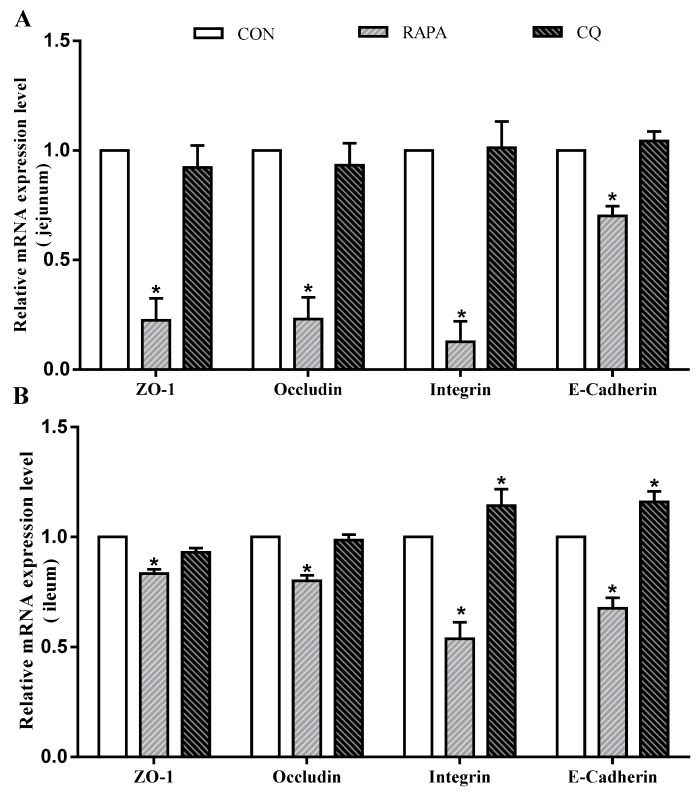
Relative mRNA levels. (**A**) *E-cadherin*, *occludin*, *ZO-1*, and *integrin* mRNA levels on jejunum; (**B**) *E-cadherin*, *occludin*, *ZO-1*, and *integrin* mRNA levels ileum. Dietary treatment: RAPA, rapamycin; CQ, chloroquine; CON, normal saline. Data are expressed as means ± SEM (*n* = 6), all values are expressed in relation to the control. * *p* < 0.05 vs. control.

**Figure 4 animals-10-00290-f004:**
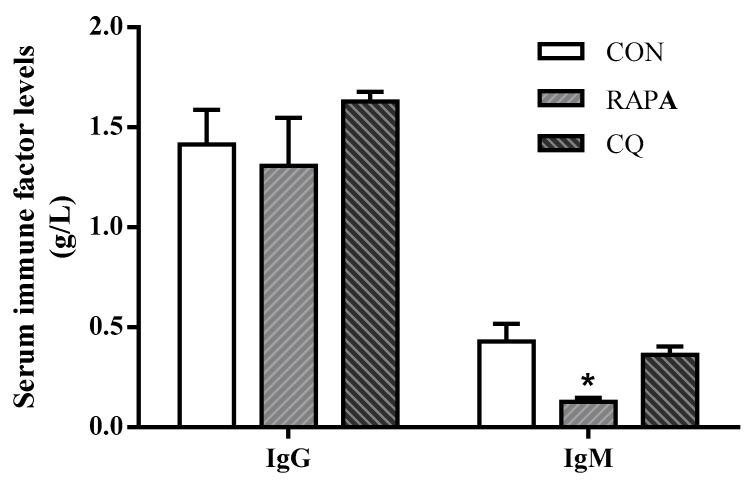
Serum immune factors levels of IgG and IgM. Dietary treatment: RAPA, rapamycin; CQ, chloroquine; CON, normal saline. Data are expressed as means ± SEM (*n* = 6), all values are expressed in relation to the control. * *p* < 0.05 vs. control.

**Table 1 animals-10-00290-t001:** Primers used for real-time quantitative PCR.

Genes	Accession No.	Primers	Sequences (5′-3′)
*E-cadherin*	NM_001163060.1	Forward	GAAGGAGGTGGAGAAGAGGAC
		Reverse	AGAGTCATAAGGTGGGGCAGT
*Occludin*	NM_001163647.2	Forward	AGAGTCATAAGGTGGGGCAGT
		Reverse	CGCCCGTCGTGTAGTCTGTC
*ZO-1*	XM_005659811.1	Forward	TACCCTGCGGCTGGAAGA
		Reverse	GGACGGGACCTGCTCATAACT
*Integrin*	NM_213968.1	Forward	GCAGTTTCAAGGTCAAGATGG
		Reverse	AGCAGGAGGAAGATGAGCAG
*β-actin*	XM_003124280.3	Forward	GGATGCAGAAGGAGATCACG
		Reverse	ATCTGCTGGAAGGTGGACAG

ZO-1, zonula occludens-1.

**Table 2 animals-10-00290-t002:** Growth performance of piglets.

Items	Dietary Treatment			SEM	*p*-Value
	CON	RAPA	CQ		
Day 1 body weight, kg	7.14	6.78	6.86	0.15	0.63
Day 7 body weight, kg	8.04	6.78 *	7.84	0.20	<0.01
Day 14 body weight, kg	9.57	6.80 *	9.99	0.40	<0.01
Average daily gain, g	173.43	1.19 *	223.29 *	26.13	<0.01
Average daily feed intake, g	261.67	122.50	254.50	16.53	<0.01
Feed efficiency, g gain/g feed	0.60	0.13	0.91	0.11	<0.01

Growth performance of piglets. Dietary treatment: RAPA, rapamycin; CQ, chloroquine; CON, normal saline. All values are expressed in relation to the control. * *p* < 0.05 vs. control (*n* = 6).

**Table 3 animals-10-00290-t003:** Serum levels of serum diamine oxidase (DAO) and D-lactate.

Items	Dietary Treatment			SEM	*p*-Value
	CON	RAPA	CQ		
DAO, mmol/L	1.40	0.72 *	1.35	0.22	< 0.01
D-lactate, μg/mL	77.75	95.67 *	63.53 *	9.30	< 0.01

Dietary treatment: RAPA, rapamycin; CQ, chloroquine; CON, normal saline. All values are expressed in relation to the control. * *p* < 0.05 vs. control (*n* = 6).

**Table 4 animals-10-00290-t004:** Serum inflammatory factors levels (pg/mL).

Items	Dietary Treatment			SEM	*p*-Value
	CON	RAPA	CQ		
IL-6	669.74	705.83 *	565.70 *	42.01	0.04
IL-12	106.20	123.07 *	104.06	24.60	<0.01
IL-1β	339.61	376.88 *	231.75 *	44.53	<0.01
TNF-α	110.68	125.06	96.10	8.53	<0.01
IL-8	38.48	44.40 *	23.56 *	6.20	<0.01
IFN-γ	25.59	27.19	13.34 *	4.37	<0.01
TGF-β	454.20	216.89 *	549.37 *	47.00	<0.01
IL-10	107.62	91.83	106.80	5.13	0.20

Dietary treatment: RAPA, rapamycin; CQ, chloroquine; CON, normal saline. All values are expressed in relation to the control. * *p* < 0.05 vs. control (*n* = 6).

**Table 5 animals-10-00290-t005:** Plasma antioxidant index.

Items	Dietary treatment			SEM	*p*-Value
	CON	RAPA	CQ		
SOD, U/ml	74.38	64.28 *	71.86	1.17	< 0.01
MDA, nmol/ml	3.65	4.56 *	3.43	0.16	< 0.01
GST, U/ml	34.32	26.61 *	42.60 *	1.72	< 0.01
GSH-PX, U/ml	105.39	86.33 *	114.52 *	3.07	< 0.01
T-AOC, mmol/L	0.22	0.18	0.22	0.01	0.02

Superoxide dismutase, SOD; malondialdehyde, MDA; glutathione S-transferase, GST; glutathione peroxidase, GSH-Px; total antioxidant capacity, T-AOC. Dietary treatment: RAPA, rapamycin; CQ, chloroquine; CON, normal saline. All values are expressed in relation to the control. * *p* < 0.05 vs. control (*n* = 6).

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
