# Peer review of "Chloroquine Downregulation of Intestinal Autophagy to Alleviate Biological Stress in Early-Weaned Piglets"

_animals, 2020, doi:10.3390/ani10020290_

Round 1
Reviewer 1 Report
The revisions you made have enhanced this manuscript. There is still a moderate number of English grammar and spelling errors throughout the manuscript that need to be revised. The introduction is more clear and the reader now understands why you did this study. The entire experimental period and timing is still not clear. I suggest a figure using a timeline to demonstrate your study rather than trying to again explain and write this in English as that seems rather difficult for you.
The diet appears to have changed. Now you use zinc sulfate at a lower dose, whereas before you had a high dose of ZnO - what happened? This does not give confidence in the quality of the research.
Finally, quantification of changes to responses is missing from the results section. It is important to quantify at least some results to give the reader an idea of how large or how small the changes are. This is because the change can be significant (P < 0.05) even though it is only a 10 g difference and that 10 g difference, even though significant, is not commercially relevant.
L28: ‘additive’ not addition
L39-40: Indicate which treatment you are compared with CQ? RAPA or control? “The CQ-treated piglets showed higher final BW and ADG, improved villus 39 height in the jejunum and ileum, and lower autophagy and inflammation (P < 0.05).”
L87: I still do not like how you present age and time. I would leave this out for the introduction and just say, “in pigs post-weaning, to assess their…”
Introduction: The additions of RAPA and CQ are excellent and well explained. Now, I (and the reader) clearly understand the model you chose, why you chose it, and what you expect from it.
L95: 24 days (d) of age.
L97: 1 mg RAPA / kg BW was the dose. Please indicate the total volume of solution that was orally administered as well for saline, rapa, and CQ treatments.
Section 2.1. is still not clear. You need to outline all steps day by day so the reader can understand your method. Perhaps a figure of an experimental timeline would be more helpful.
L102: Now you have ZnSO4 in the premix and not ZnO. Why did the premix change since the first version? I also think the diet has changed since the first version to the same as the diet you reference. I am now confused. The diet is more standard now though and the ZnO is not a problem…
L108: Instead of writing “when the feed trial ended” you should write “On d 41 of age, piglets were euthanized.
Section 2.9: Results are not expressed as the mean ± SEM in tables. Furthremore, you need to indicate in this section that you only did 1 way anovas that compared control vs. rapa and control vs. CQ and that you never statistically compared RAPA and CQ.
Section 3. Results: I still miss quantification of results in all sections. For example you could write, RAPA decreased (P < 0.05) final BW by 2.77 kg, whereas CQ increased (P < 0.05) final BW by 0.42 kg compared with control pigs. You can also use percent increases and other methods to quantify the changes you observed. It is important to do this so the reader can quickly understand the impact of your treatments. RAPA had a huge impact on performance whereas CQ had a minimal impact, but was good at maintaining conditions similar to control.
Table 3: G:F instead of feed efficiency
Author Response
Dear Reviewer,
Thank you for your comments on our manuscript entitled “Chloroquine downregulation of autophagy to alleviate biological stress in early-weaned piglets” (ID: animals-702279). We have carefully checked the manuscript again. We would like to express our great appreciation to you for comments on our paper. Responds to the comments are as attached.
Best regards,

Reviewer 2 Report
General comments
Most of the comments have been considered, however something is still missing. In general, attention should be used in the use of commas. Moreover, English should be revise: lots of verbs are not in the correct form (especially in the section “Materials and methods”). Please, note that the passive form should be used (e.g. line 97: “piglets were treated by oral administration daily and…” and not “piglets treated by oral administration daily”; line 99: “Every single pig housed in a cage equipped with a feeder”; line 122 and so on). The style of the text must be formatted and should be the same for all the manuscript, also for the parts that have been recently added.
Introduction
Line 53 (previously: line 49): the comment has not been considered. What do you mean with “which is considerably less so”? Please, revise the sentence
Line 80: regarding the new part added, the use of “DDS-induced mice” should be clarify. Please use the entire form (dextrane sulfate sodium) and then the abbreviation.
Materials and methods
Line 97: Please, use “once a day” or “daily”. “once daily” is not a correct form.
Line 98: The sentence “to adjust…the body weight” is not clear. Please, clarify the sentence.
Transmission electron microscope
Line 128: please, the subject is missing. You can write “we cut the jejunal tissue” or “the jejunal tissue was cut”
Conclusion
The conclusion chapter is still missing. A separate paragraph called “conclusions”, as for “results” and “discussion”, has to be added including the most relevant findings of this research.
Table 1
The superscript “1” that appears under the table is not present in the table itself. What is it referred to? Maybe is the antioxidant composition?
Table 3
Line 448: “Growth performance of piglets. Body weight, average daily gain, average daily feed intake, Gain/feed.” this line is not necessary as you don’t use abbreviations in this table, so it is not necessary to rewrite them under the table.
Author Response

(The authors gave the same response as above.)

Reviewer 3 Report
The authors did not include one of my comments.
Point 3: Figure 1 A - I am a supporter of showing the obtained values in tabular form - the graphs presented are difficult to read.
Responds: Revised as requested.
Authors should place on the graph bars (Figure 1-4) the values regarding the parameters in individual groups.
Other comments have been supplemented.
Author Response
Dear Reviewer,
Thank you for your comments on our manuscript entitled “Chloroquine downregulation of autophagy to alleviate biological stress in early-weaned piglets” (ID: animals-702279). We have carefully checked the manuscript again. We would like to express our great appreciation to you for comments on our paper. Responds to the comments are as attached.
Best regards,

This manuscript is a resubmission of an earlier submission. The following is a list of the peer review reports and author responses from that submission.
Round 1
Reviewer 1 Report
The use of commas and English grammar is poor. I started revising it, but there is so many grammatical errors that I stopped to revise it for these mistakes. Therefore, major grammar revisions are necessary prior to acceptance for publication. Furthermore, it is not clear why you would hypothesize that RAPA and CQ would produce similar effects in piglets, but in reality, they produced complete opposite effects. You need to more clearly specify why you tested both compounds and what the difference between the 2 could be from the start. Only providing citations is not enough. Also, I believe it is impossible for pigs to consume 150 g of feed per day and for them to not gain any weight over 14 d. Either the dose of RAPA was quite serious or something is wrong with your calculations. Or those pigs wasted a lot of feed. This data is hard to believe. You definitely need to quantify this in your results section as well as try to explain this in the discussion or it is hard to accept these results. Also, you could further try to suggest that RAPA would be a good agent to use a model of impaired intestinal barrier function and extreme post-weaning stress. What would you expect to happen if you fed CQ to RAPA fed pigs?
Title: Can you be more specific with what or where the autophagy is? Perhaps intestinal autophagy?
L23: '.' after weaning
L23: delete the ',' after CQ
L24: delete the ',' after barrier
L25: to find potential value for 'CQ' as a feed 'additive'
L28: change anticipated to 'hypothesized'
L31: 'or' treated with
L31: delete ',' after autophagy
Abstract: need P-values
L54-56: delete last sentence
L60: delete ‘have’
L63: delete ‘diverse’
L66-69: example to obesity is not relevant
L73: ‘R’ should be ‘RAPA is’
L79: 14-day ‘old’, early ‘weaned’ piglets and delete ‘,’ after piglets
L82: experiments ‘were’ conducted
L86: you said 14-day old early weaning piglets in L79, but here you say 24 d of age so what is it? The initial BW indicates that it is the latter.
L86: delete ‘,’ after age)
L87: (BW) ‘of 7.03 ± 0.57 kg) and delete ‘,’ after kg) and delete ‘adopted and’
L88: Replace ‘, and’ with ‘. Pigs from each treatment group were provided an equal..’
L89: delete ‘after being weaned’
L93: what was the feeding level? Why did you adapt pigs for 3 days? The experiment lasted from d 24 until d 38 of age? Pigs then were euthanized on d 38 of age (d 14 of study)? All of this needs to be clarified.
L98: replace ‘killed’ with ‘euthanized’
L99: replace ‘slaughtered’ with ‘euthanized’ and throughout paper
Results: need to quantify the changes and add P-values throughout the results section.
L229: increased intestinal permeability is impaired barrier function. Neither are a result of each other.
L242: introduce CQ in introduction using this as your hypothesis.
Figure 1: How could pigs fed RAPA not gain BW at all even though they are eating 150 g of feed per d? This is wrong or needs to be explained in results and discussion. This is why you need to quantify your results in the results section.
Table 1: What is corn for Suckling pig? This is a terrible and non-relevant diet to feed to a weanling pig. The high inclusion of soybean meal is a huge challenge to these pigs. Is there a reason you used this diet and this high level of soybean meal? You need to explain this. Also, you used a high level of ZnO – you need to mention this and you also need to discuss your results with this high level in mind.
Table 5: The superscripts are incorrect according to your SEM and your footnote. Especially for GST.
Author Response
Dear Reviewer,
Thank you for your letter and for the comments concerning our manuscript entitled “Chloroquine downregulation of autophagy to alleviate biological stress in early-weaned piglets” (ID: animals-648806). Those comments are all valuable and very helpful for revising and improving our paper, as well as the important guiding significance to our researches. We have studied comments carefully and have made correction which we hope meet with approval. Revised portion are marked in red in the paper. The main corrections in the paper and the responds to the reviewer’s comments are as attachment:

Reviewer 2 Report
Review A very interesting manuscript, raising critical points in pig farming. Some comments:In the introduction, authors should devote more attention to experimental factors RAPA and CQ.
Lines 88-89 - The experimental setup should be clearer.
Lines 89 - RAPA or Rapa? - standardize
Line 88-89 - the authors write "saline (8 ml), Rapa (1 mg / ml) [22], and 88 CQ (10 mg / ml BW)" - how often animals were weighed to adjust the dose of preparations to body weight ? In addition, the authors refer to the literature on the doses used - I believe that this should be explained here by the amount of such factors introduced (RAPA and CQ) as well as saline.
Lines 89 - the authors write "by oral administration once daily after being weaned." - the question arises whether this form of administration of the agent is not too stressful for animals? Could this not affect the test result?
Figure 1 A - I am a supporter of showing the obtained values ​​in tabular form - the graphs presented are difficult to read.
Table 4 - the unit should be in the title of the work.
I believe that the discussion is correct, however, the authors should develop the discussion towards explaining the lead-like mechanisms that caused such a reaction. Authors should base on references.
To sum up, the manuscript is very interesting, based on a very wide range of analyzes and providing interesting information that encourages further research in this direction.
Author Response

(The authors gave the same response as above.)

Reviewer 3 Report
MANUSCRIPT ID: Animals-648806
The manuscript entitled “Chloroquine downregulation of autophagy to alleviate biological stress in early-weaned piglets” is focused on the evaluation of supplementation of rapamycin and chloroquine for ameliorating autophagy catabolic process in intestinal barrier and inflammation of early weaned piglets.
The study was developed on an interesting and original topic: the interest in autophagy processes of intestinal barrier, the first line of defence against bacterial invasion, has exploded over the last decade, with publications highlighting crosstalk with several other cellular processes. Moreover in the intensive pig production the weaning represents the most critical phase where several factors can affect animal health, performance and the profitability of the livestock.
However revisions are required: some suggestions are detailed below for the improvement of the manuscript.
In general, the references are not organized according to the style of Animals guidelines. Please, pay attention to the punctuation and the use of commas: e.g.: line 43 after “environmental”, line 54 after “failures”, line 83 after “protocols”; line 59: the sentences do not need commas.
Simple summary
Line 23: Write the acronyms with the extended form, at the first time in the text all acronyms should be written in the complete form.
Line 24: Remove the comma after “intestinal barrier”.
Abstract
Line 27: please substitute “food intake” with “feed intake”.
Line 30: Please clarify what do you mean with “of 6” .
Introduction
Line 44: Substitute “food intake” with “feed intake”.
Line 49: what do you mean with“which is considerably less so”?
Line 59: Please provide a reference. Are you sure you can speak about upregulation?
Line 60-61: did you published these data? Which kind of toxin did you used?
Line 72: Please, uniform the use of abbreviations in the entire manuscript. As example: RAPA (all in capital letter) or “Rapa” (line 89)? The same should be done for chloroquine, “CQ” or QC (Table 4).
Line 73-74: what do you mean with “R”? and “LC3”?.
Materials and Methods
Animals and Experimental treatments
Line 82: please check the English (were conducted?!). Please provide the ethical authorization number.
Line 89: please explain better the administration of the treatments. Did you administer the treatments by gavage?
Line 90-93: please revise the sentence. It is not clear? What do you mean with “adoption”?Are they housed in individual pen? What do you mean with environment? Is it a room? Please explain better the experimental design.
Line 92: Please, update the reference of NRC.
Sample collection and preparation
This part is not in formatted style required from the journal, please revise it.
Line 99-100: why did you speak about “slaughtered” if they are euthanized? Please describe the collection of blood samples (vein, needles…).
Line 102-103: Can you be more precise on the sample storage? The sentence must not start with “And”. Avoid the use of “some”, indicate a precise amount. Please, revise the sentence.
Western blotting analysis
Line 107: Write all the acronyms in the extended form at the first time that they appear in the text.
Line 110-113: please revise the sentence. The correct verbs are lacking.
Transmission electron microscope
Line 116-120: Revise the sentence. It is not clear.
Line 117: Please use superscript for referring to mm3, remove the “a” before “2.5%” or add “solution” after glutaraldehyde.
Serum inflammatory cytokines, immune factors, diamine oxidase and D-lactate
Line 126: “IGG and IGM” should be written as “IgG and IgM”. Please, revise them also in the tables section.
Line 133: which kind of assay kits are you talking about? ELISA? Please provide more information about the kits.
Intestinal morphology
Line 141-142: Correct the typos and revise the English.
Real-time quantitative reverse transcriptase PCR
Line 145-162: Please, provide more information about the PCR conditions used.
Line 145: Correct the typo.
Line 150-151: Check the punctation.
Statistical analysis
Line 167: Did you consider 0.05 as significant? If yes, please include it in the statistical significance P ≤ 0.05 otherwise in the statistical tendency ≤ 0.05 P < 0.1
Results
Include the p-values in the text between brackets when you refer to a statistical difference.
Intestinal morphology
Line 184: Remove the dot after “Figure 3.”.
Line 186: Please clarify the sentence explaining better what you refer to for “lower levels when compared to control group”.
Line 188: In way you explain the association between Intestinal permeability and the levels of tight junction proteins? Is there a statistical correlation? Add the space after “proteins” before “(Figure 4.)”. Remove the dot after “Figure 4.”.
Discussion
Line 209: Use the italic for in vivo and in vitro in the entire manuscript.
Line 220: Add a space between “indicators” and the reference.
Line 221: Correct the typo.
Line 222-223: please revise the sentence: it is not clear.
Line 235: What do you mean with “exaggerated oxidative damage”?
Line 236-238: The sentence is not clear, please revise it.
Conclusion
The conclusion chapter of the manuscript is missing, please add a general statement that includes the most relevant findings of this research.
References
Please revise the references according to the journal guidelines.
Table 1:
List the ingredients in descending order from the ingredient included at the highest concentration to the lowest.
Please explain what you mean with “Corn for Suckling pig”, “second head”, and “piglets multidimensional”.
Provide more information about which antioxidants are included in the feed ingredient “antioxidants”.
Regarding the diet, can you supply the calculated or analysed content of nutrients (dry matter, crude protein, crude fiber, ether extract, ashes, digestible energy)? Are you sure that it is balanced according NRC?
Table 3-4-5: these tables are not self-explanatory, please fix Table 3 and 4 in the correct position (now they are not readable).
Author Response

(The authors gave the same response as above.)
